# Corneal Healing and Recovery of Ocular Crystallinity with a Dichloromethane Extract of *Sedum dendroideum* D.C. in a Novel Murine Model of Ocular Pterygium

**DOI:** 10.3390/molecules26154502

**Published:** 2021-07-26

**Authors:** Luiselva Torrescano-De Labra, Enrique Jiménez-Ferrer, Brenda Hildeliza Camacho-Díaz, Gabriela Vargas-Villa, Manases González-Cortazar, Maribel Herrera-Ruiz, Sandra Victoria Ávila Reyes, Javier Solorza-Feria, Antonio Ruperto Jiménez-Aparicio

**Affiliations:** 1Centro de Investigación Biomédica del Sur (IMSS), Argentina No. 1, Col Centro, Xochitepec C.P. 62790, Morelos, Mexico; luiselva.5@hotmail.com (L.T.-D.L.); vagy@live.com.mx (G.V.-V.); gmanases@hotmail.com (M.G.-C.); edanae10@yahoo.com.mx (M.H.-R.); 2Centro de Desarrollo de Productos Bióticos del Instituto Politécnico Nacional, Carretera Yautepec-Jojutla, Km. 6, Calle CEPROBI No. 8, Col. San Isidro, Yautepec C.P. 62731, Morelos, Mexico; bcamacho@ipn.mx (B.H.C.-D.); jsolorza@ipn.mx (J.S.-F.); 3CONACyT—Centro de Desarrollo de Productos Bióticos del Instituto Politécnico Nacional, Carretera Yautepec-Jojutla, Km. 6, Calle CEPROBI, No. 8, Col. San Isidro, Yautepec C.P. 62731, Morelos, Mexico; sandra_victory@yahoo.com

**Keywords:** *Sedum dendroideum* D.C., pterygium, anti-inflammatory, scar process

## Abstract

Pterygium is a corneal alteration that can cause visual impairment, which has been traditionally treated with the sap of *Sedum dendroideum* D.C. The pharmacological effect of a dichloromethane extract of S. dendroideum was demonstrated and implemented in a pterygium model on the healing process of corneal damage caused by phorbol esters. In mice of the ICR strain, a corneal lesion was caused by intravitreal injection of tetradecanoylphorbol acetate (TPA). The evolution of the corneal scarring process was monitored with vehicle, dexamethasone, and dichloromethane extract of *S. dendroideum* treatments by daily ophthalmic administration for fifteen days. The lesions were evaluated in situ with highlighted images of fluorescence of the lesions. Following treatment levels in eyeballs of IL-1α, TNF-α, and IL-10 cytokines were measured. The effective dose of TPA to produce a pterygium-like lesion was determined. The follow-up of the evolution of the scarring process allowed us to define that the treatment with *S. dendroideum* improved the experimental pterygium and had an immunomodulatory effect by decreasing TNF-α, IL-1α, and maintaining the level of IL-10 expression, without difference with respect to the healthy control. Traditional medical use of *S. dendroideum* sap to treat pterygium is fully justified by its compound composition.

## 1. Introduction

Pterygium is popularly known in Mexico as “carnosidad” (“fleshiness”) in reference to its resemblance to a flesh formation on the surface of the cornea. It has been defined as a proliferative fibrovascular condition on the ocular surface, and its etiology is not fully understood, but it is a public health problem [1]. Approximately 200 million people worldwide suffer from this condition, which in severe cases can affect vision by leading to the development of astigmatism or physical occlusion of the visual axis [2]. The triangular aspect resembles the wing of an insect due to the high degree of vascularization and is very similar to an aberrant wound healing response associated with a prolonged recovery process [3]. Its prevalence is higher in rural areas than in urban areas, and its incidence is highest in the population group between 40 and 50 years of age, affecting both sexes equally. Its development has been observed in people who work in open outdoor spaces, where there is greater exposure to factors such as dust and sunlight [4]. Among the main elements and signs that characterize it are conjunctival congestion, redness, edema, and opacity of the cornea, the latter described as “spots on the cornea” [5].

In traditional Mexican medicine, in the states of Morelos and Guerrero, the sap of “Siempreviva amarilla” (*Sedum dendroideum* D.C.) is recommended as a treatment for pterygium, applied by directly squeezing the leaf into the eyes [6,7]. This plant from the Crassulacea family has fleshy, green, and spatula-shaped leaves and its petals look like stars and are bright yellow. The plant is also suggested to treat toothaches, smooth tooth surfaces, whiten teeth, and cure “cooked mouth” and periodontal disease. It is also applied to the eyes as a treatment for age-related cataracts. The main conventional treatment for pterygium is surgical removal, however, since the condition itself is an alteration of the scarring process, recovery from surgery results in many recurrences [8].

The mammalian cornea is made up of three layers: a stratified outer layer, an intermediate stratum that corresponds to the stroma, which is a thick, relatively inactive layer of collagen that is constantly remodeling, and an inner layer of endothelial cells. After a corneal injury, the immediate, short-term response is a process of rapid closure of the injury with a significant increase in the level of MMP-9. The long-term response is dependent on the activity of fibroblasts that reside in the corneal stroma and the activity of MMP-1, which slowly remodels the site of injury [9]. In the healthy cornea, this wound repair is finely regulated and contrasts with what is observed in the development of pterygium, where the main alteration is associated with the fact that enzymatic activity is not controlled by specific MMP inhibitors [10].

There is evidence that PMA (phorbol myristate acetate) causes a positive feedback process for collagenase production, mediated by IL-1α [11]. The increase in the level of collagenase prevents excessive collagen deposition, but if the level of collagenase is excessive, destruction, tissue remodeling, and organic damage can occur that prevents the culmination of the scarring process due to a persistent inflammatory signal. In some reports of pterygium, when determining MMP-1 and its inhibitor TIMP-1, the latter was observed at levels similar to those of MMP-1, which could explain the progressive-invasive growth of the pterygium [10]. Fibroblasts close to the lesion secrete IL-1α in response to IL-1 secreted by macrophages responsible for the inflammatory process while migrating to fill and give consistency to the damaged area. Fibroblasts found within the stroma and far from the injury site are exposed, and being unable to secrete IL-1α in response to IL-1, they stop the cycle. In pterygium, the persistent erosion of cells, together with excessive collagenase, maintains the remodeling process, and cells that migrate to cover the lesion surface secrete MMP-9, and the inflammatory process continues [9]. Other cytokines are involved in this model, such as IL-10 and TGF-β, which acts as an antagonist of IL-1α [12].

The pharmacological effect of *S. dendroideum* in the process of corneal scar formation in a pterygium model was evaluated, and the follow-up of the evolution of the scarring process, allowed to define that the treatment with *S. dendroideum* had an immunomodulatory effect by decreasing TNF-α, IL-1α, and maintaining the level of IL-10 expression, without difference with respect to the healthy control, demonstrating that traditional medical use of *S. dendroideum* sap to treat pterygium is fully justified by its compound composition.

## 2. Results

### 2.1. Phytochemical Analysis of Dichloromethane from S. dendroideum Lyophilized Leaf Juice

Approximately 2.5 L of fresh leaf juice was extracted from 2.9 kg of fresh leaves which resulted in 69 g of powder. From this, 736 mg of dichloromethane extract was extracted. GC-MS analysis of the dichloromethane extract revealed striated fatty acids, policosanols, and isoprenoids. Several compounds were not detected probably due to their low concentrations. Those that were identified in the dichloromethane extract in the three fractions that were analyzed by GC-MS were: 12-Triaconten-1-ol which represents 25.25% of the total extract (Figure 1), β-Sitosterol, α-tocopherol, α-amiryn, and methyl octacosanoate, and in a lower percentage, phytol, gedunin, hexacosanol, octacosanoic acid methyl ester, α-tocopherol, ciclooctacosane, α-amyrin, and 1-Dotriacontanol.

### 2.2. Injury Model for Pterygium

#### 2.2.1. Dose Effect Curve

The intravitreal administration of TPA caused the appearance of a corneal lesion, which was evidenced by the fluorescence emitted by the applied reagent and exposure to ultraviolet light. The measurement of the lesion area was expressed in pixels, and they were measured at Four different times after administration of the TPA challenge at 3, 7, 10, and 15 days. The initial size of the lesion followed a dose-dependent trend, and the lesions healed (i.e., decreased in size) over time (Figure 2). With a dose of 0.156 mg, the lesion caused by TPA spontaneously remitted at the end of the measurement period. The evolution over time of the size of the lesion resulting from administration of 0.312 mg TPA was described by an exponential decay curve with a decay rate constant 28% higher than that of the lowest dose of TPA administered. The average decay rate among the six trials testing the 0.312 mg dose was −0.067 pixels/day, showing the reproducibility of results, and at the end of the evaluation period, the lesion was still measurable. The lesion decreased in area by 55% relative to the initial size.

At a dose of 0.625 mg, the lesion’s average size decrease was 0.019 pixels/day, with a final decrease of 21% relative to the initial size. Although, it should be noted that the size of the lesions generated by the 0.612 mg were two orders of magnitude larger than those obtained with the 0.312 mg dose, which was not adequate, this size of lesions did not allow an evaluation of the pharmacological activity of the treatments that were proposed.

#### 2.2.2. Corneal Injury Induction by TPA Administration and Treatment

Figure 3 shows the comparison of the evolution kinetics of the corneal lesion when exposed to three types of treatment. To evaluate the evolution of the lesions with the treatments, the first group of mice received only TPA (negative control), the next group received TPA plus treatment with dexamethasone (positive control), and the experimental treatment received TPA plus dichloromethane extract (TxExp). All these were compared against the control group that did not receive TPA, and only the vehicle that consisted of a mixture of water: acetone. In all cases, the development of the lesion followed an exponential decay model. The equations that described the behaviors were: y = 2 × 10^6^e^−0.174^ group negative control, y = 6.57 × 10^5^e^−0.1^ group positive control, y = 3.45 × 10^5^e^−0.171^ group of vehicle, and y = 4.35 × 10^5^e^−0.108^ the TxExp. It is noteworthy that on the third day of follow-up of the evolution of the size of the lesion, there were significant differences. When comparing the size of the lesion between the negative control group and the dexamethasone treatment, it decreased by 67.15%, and regarding the experimental treatment, the decrease was 78.25%. On the other hand, when comparing the recovery rates of the lesions, the treatments with the steroidal anti-inflammatory drug and with the dichloromethane extract were around 40% slower in both cases, with respect to the decay curve of the negative control.

#### 2.2.3. Injury Induction by TPA Administration and DIA

When analyzing the size and shape of the lesion (Table 1), it is again observed that in the three treatments, the lesion tends to remit spontaneously, and treatment with dexamethasone and with the dichloromethane extract of *S. dendroideum* decreased the relative area of damage, which relates the affected area to the total area analyzed. In the evolution size of the lesion, a significant difference was presented in the treatment with dexamethasone, with a decrease of 57% between the final time and the onset of the lesion. In contrast, the positive control treatments and the dichloromethane extract of *S. dendroideum* did not show a significant decrease in the diameter of the lesion. In the process of scar, significant differences were observed between the initial and final states of the three treatments. When comparing the treatments against the damaged group, significant differences were observed in lesion perimeter, on the initial and final state, both in dexamethasone and extract treatment groups. The Feret’s diameter was different only in its final state in the experimental treatment group.

To analyze the evolution of the healing process, Figure 4 shows the analysis of the scar irregularities. For this, the fractal dimension of the area was considered, where the degree of regularity corresponds to the value approximation = 2.0 (AFD), and of the perimeter, where the greatest regularity is when the value of PFD approaches the value = 1.0.

Panel (a) shows the evolution of the scarring process when the negative control treatment was applied. It can be seen that with respect to AFD, the lesion gradually moves away from the limit value for this parameter, following a trend that can be adjusted to an equation of the line y = −0.0073(x) + 1.91. In the same way, a departure from the limit value for PFD is observed, although it is much slower (two orders of magnitude) y = 0.00002(x) + 1.35.

Panel (b) shows the scar process with dexamethasone treatment, where the trend of the healing evolution does not change, although the AFD value slows down since the slope of the equation of the line decreases its change value by half (y = −0.004(x) + 1.91). On the other hand, with respect to PFD, the change was 50 times faster than the negative control (y = 0.001(x) + 1.32). In the experimental treatment (panel c), as with dexamethasone, the trend did not change with respect to AFD, although the process slowed down even more since the rate was −0.003 (y = −0.003(x) + 1.88). The most important modification was observed with respect to the PFD since the rate of change of this parameter is 200 times higher (y = 0.004(x) + 1.275).

Figure 5 shows the corneal opacity values using the luminosity in the red channel, where a mouse eye with greater crystallinity is dark red. Thus, higher values of red luminosity correspond to a healthy organ, and eyes with a degree of corneal opacity as a consequence of the scar, tend to pink tones. Opacity is the deleterious condition and transparency or crystallinity is the condition of a healthy eye. Thus, the transition between opacity and luminosity (0–255) of the red channel corresponds to the recovery of the basal or healthy ocular condition. Regarding opacity, the administration of TPA caused an increase in opacity (OA = 38.5) of 1.16 times compared to the OA of the baseline group (OA = 17.75). The treatment with dexamethasone decreased (OA = 33.0) the opacity to 0.86 times and the experimental treatment decreased (OA = 28.8) the opacity to 0.62 times, when comparing against the baseline control group.

When comparing the opacity with the area of injury, it was observed that in TxExp, the opacity values indicated greater crystallinity of the eyes of that group compared to the Dex and NegCtrl groups, with a decreasing area of damage indicating recovery. When determining the index that relates both variables, of the three groups, it is lower in the group of the experimental treatment TxExp. For the angiogenesis index, when comparing against the basal group, a significant increase of this index could be observed in the damage group with TPA, with a 2.51 times increase (IA = 2.32). Treatment with dexamethasone decreased this value to 2.33 times (IA = 2.20), with a greater reduction when administering the experimental treatment with 0.63 times (IA = 1.08) (Figure 6).

### 2.3. Histological Analysis

Gomori’s trichrome staining marks the difference in the blue color of the accumulation of collagen [13], which in the case of this work is a consequence of the induction of the scarring inflammatory process, characteristic and closely related to pterygium. Muscle and cytoplasm structures stain is represented in red or pink, and cell nuclei stain in black. In the representative micrographs of the groups, a portion of the cornea is presented, where three layers are observed (Figure 7).

The photomicrograph allows us to identify an ordered stratification of the layers of a healthy cornea of group HC (panel a). In this, in order from the outside to the inside, the outer layer or corneal epithelium was identified, followed by the intermediate layer or stroma, where both are the same thickness, towards the inside, the endothelium, and finally, the inner layer of the cornea, which corresponds to a fine tinted red.

The damage caused by TPA, which corresponds to the negative control of the pterygium model, presents severe alterations of the cornea (panel b). The micrograph shows that the corneal epithelium decreased in thickness with respect to the stroma and presents an apparent decrease in the density of the cell population. One of the most severe damages is related to the opening of spaces between cells and the detachment of some of these. The stroma also shows cell shedding and the endothelium disappears, causing apparent thinning of the cornea in this group. Treatment with dexamethasone caused a gradual recovery of the cornea showing a tissue organization similar to that observed in the healthy HC group (panel c). However, there is detachment of the epithelial layer and apparent opening of spaces in the stroma stain blue, corresponding to collagen from the scarring process.

In the TxExp group, it is evident that the administration of the extract led to the maintenance of the corneal layers (panel d). However, it presented cell detachment both in the epithelium and in the endothelium and an apparent increase in the density of the cells is observed. The appearance of red spots on the epithelial surface was evidenced by the possible induction of cell proliferation. During the healing associated with the treatment with the dichloromethane extract, the epithelium shows evident detachment, it is still present, unlike what was observed in (panel b) where it was completely degraded.

### 2.4. Cytokine Level in Eyes

Cytokine determination was performed in the tissue homogenates of the samples from the Veh1 and Veh2 groups, which only received the dissolution vehicles TPA and extract, without TPA and maintained the ocular concentrations of TNF-α close to the healthy control group (HC). The concentration of TNF-α proinflammatory cytokine in NegCtrl was approximately three times higher than HC. In Dex, the concentration of TNF-α was close to the HC group. Finally, in the TxExp group, the concentration of TNF-α was lower than in the HC group (Figure 8).

Similarly for IL-1β, the Veh1 and Veh2 vehicle groups (no lesions) maintained ocular concentrations of TNF-α close to the HC group, although the administration of acetone increased the concentration of IL-1β compared to the group receiving cyclodextrin and Tween. The concentration of IL-1β in NegCtrl was approximately twice that of the HC group. For Dex, the concentration of IL-1β was similar to Veh1 and HC groups. TxExp showed a decrease in IL-1β concentration, though this decrease was not as significant as the decrease in TNF-α (Figure 9).

The effects on IL-10 were opposite to those on TNF-α and IL-1β, which was as expected since IL-10 is an anti-inflammatory cytokine. The HC group showed high concentrations of IL-10. The Veh1 and Veh2 vehicle groups maintained ocular concentrations of IL-10 close to the HC group. The administration of acetone induced a decrease in levels of IL-10, but not as drastically as in NegCtrl. The concentration of IL-10 in NegCtrl decreased to approximately half the level of HC. In Dex, the concentration of IL-10 remained close to the HC group. TxExp showed an increase in IL-10 concentration (Figure 10).

## 3. Discussion

An experimental injury model for pterygium-like lesions was established in mice, unlike previous studies carried out in rabbits, where a type of neoplasia was generated as a result of ultraviolet light [14]. While this work pretends to be a model of pterygium, it is important to define that the difference with the human pterygium is the absence of scar tissue on the cornea, this due to the structure of the mouse eye, where the proliferation of fibroblasts and connective tissue does not occur as in humans [10,15]. However, there is a greater proliferation and cell density in the injured cornea, compared to a healthy eye, as occurs in the human pterygium [16], and this was verified with digital analysis using the LSCM and ESEM tools (data not shown). We then used this model to test the extract from lyophilized leaf juice of *S. dendroideum*.

The administration of TPA in the vitreous humor caused a harmful environment that resulted in the appearance of lesions on the cornea [17]. The severe inflammatory process originated from inside the eye, at the level of the vitreous humor, subsequently damaging the lens and finally the cornea. This caused lesions that were detectable using green fluorescence, which was observed with the use of sodium fluorescein. The proposed treatments seek to recover the transparency of the lens and the correct healing of the cornea.

Dexamethasone contributed to the healing process, generating less elongated lesions, which suggests an organized scarring process, but crystallinity was not fully recovered [18]. Glucocorticoids are fat-soluble molecules that are absorbed through the mucosa very quickly, and once bioavailable, their free fraction enters the cells to exert their activity [19]. Therefore, glucocorticoids exert their effect at the nucleus level by direct transfer, binding to DNA to stimulate the transcription of genes that stimulate the synthesis of proteins that act as protease inhibitors or are involved in the synthesis of prostaglandins and leukotrienes for approximately 24 to 48 h [20]. This can inhibit transcription factors that are directly involved with collagenase-1 transcription [21].

The eyes of the mice exposed to TPA and treated with TxExp showed no difference in the area and perimeter of the lesion (Table 1) when comparing the initial state against the final condition, which in principle could indicate that the healing process continues a different evolution. This is confirmed by the change in the Feret diameter since with the NegCtrl and Dex treatments, the values went from values greater than two and decreased to values around one, which indicates that the elongated scar becomes a circular scar. This contrasts with the value of TxExp, which indicates that the closure of the scar follows a pattern, and the scar remains elongated.

This was consistent with the measurement of the relative damaged area (Figure 5). However, the evolution of the healing process was different, since the evolution of the opacity values showed a decreasing trend, which was parallel to the trend of the decay curve of the parameter of the relative damaged area. This was contrary to what happened with dexamethasone treatment, where the change in opacity had a trend that followed a trend that adjusted to an exponential growth, which increased during the evolution of the scarring process and this trend was contrary to the evolution of the values of the relative damaged area, which was fitted to an exponential decay curve. However, it would seem that the inflammatory process and healing are modulated differently, since the levels of pro-inflammatory cytokines, TNF-α and IL1β, were lower with the experimental TxExp treatment, compared with the dexamethasone treatment (Figure 6 and Figure 7) [22].

Regarding the neovascularization response of eyes with TPA damage, there was a marked difference between the NegCtrl group compared to the treatments with dexamethasone (Dex) and the experimental plant extract (TxExp). The NegCtr group had higher neovascularization, followed in descending order by Dex, TxExp and finally, no neovascularization was detected in the baseline control group (HC). This process is directed by an increase in the expression of multiple proangiogenic growth factors and cytokines, in combination with the decrease of angiogenesis inhibitors [23,24] reported the ability of one of the extract compounds, gedunin, to modulate aldose reductase (AR), phosphatidyl inositol-3-kinase (PI3K)/Akt, and pathways of nuclear factor kappa B (NF-κB) and block angiogenesis. Administration of gedunin suppressed the development of BPH carcinomas by inhibiting the PI3K/Akt and NF-κB pathways by inactivating Akt and the inhibitory kappa B kinase (IKK), respectively. Gedunin blocked angiogenesis by negatively regulating the expression of miR-21 and proangiogenic factors, vascular endothelial growth factor, and hypoxia-inducible factor 1 α (HIF-1α). Thus, the application of the extract may inhibit the expression of these factors and the induction of angiogenesis inhibitor IL-10 [25] in the TxExp group compared with the other groups. The elevation of IL-10 concentration is consistent with the results of angiogenesis.

The study of the fractal dimension helped to describe the changes in the morphology of the lesions without loss of fine details [26] since it allowed quantitative measurements of characteristics that are usually described qualitatively [27]. Azzouz et al. [28] studied the effects of daily intraperitoneal injections of hexacosanol, an active compound present in *S. dendroideum*, on nerve fiber regeneration in mice after sciatic nerve crush. Measurement of axonal regeneration by the pinch test seven days after injury showed a 40% increase in the rate of regeneration of sensory fibers in mice treated with hexacosanol compared to controls. The recovery of neuromuscular function was significantly improved, which could partly explain the scarring effect of *S. dendroideum* extract on pterygium. Morphometric analysis performed seven days after grinding showed a greater number of regenerating fibers, as well as a greater diameter and thickness of myelin in mice treated with hexacosanol, an increase in axon regeneration, sensory and motor in the injured nerve, which led to a better functional recovery.

The points of maximum irregularity are those where the dimensions of both area and perimeter were further away from their Euclidean dimensions of 2 and 1 respectively, and the healing mechanism was similar in the TxD and NegCtrl groups. In both of these, the points of maximum irregularity revealed the filling of the surface of the lesions with inflammatory cells [29], and a proliferation of epithelial cells, fibroblasts, and an accumulation of extracellular matrix. Epithelial cells in the area of an injury flatten and develop pseudopods to interact with the extracellular matrix [30]. Changes in cell organization, accumulation of collagen stained in blue, and epithelial cells were evidenced through histological analysis [13] and were consistent with AFD and PFD results. The changes in cell structure influenced the perimeter fractal dimension (PFD). The PFD presented greater sensitivity to changes during the injury closure process since it presented more contrasting changes in each of the stages, and because of degradation of epithelium, which was confirmed in histological analysis. In the case of the Dex group, irregular areas with smoother edges were observed. Measuring the points of maximum irregularity in the groups gives us an approach to understanding the behavior of the cells that proliferate during the stages of healing. This is similar to the analysis of the fractal dimension of cancer cells, where a relationship was found between the fractality of the cells and the invasiveness of the type of cancer. When the behavior of the cell growth approached the points of maximum irregularity, the cancer was more invasive [30]. As the cells in the inflammatory scar process mature, the fractal dimension moves away from the points of maximum irregularity. Bernard et al. related the fractal dimension with the maturity of oligodendrocytes, as the oligodendrocytes mature, their fractal dimension moved away from the points of maximum irregularity [31].

Likewise, the processes of contraction and expansion in lesion cells could be related to processes that occur in the growth kinetics of plant cells in cell aggregates of suspended cultures, where there is a filling in of the surface of the aggregate and subsequent expansion, which leads, depending on the stage, to the loss or increase of the roughness of the perimeter [32].

The polarity of the extract makes it able to exert its biological activity through IL-1 either by inhibiting or activating the transcription of genes important to the inflammatory process [33]. Otuki et al. reported the inhibitory activity of mouse ear edema by α-amyrin, a constituent of the ethereal extract of *Protium kleinii* (Burseraceae), a medicinal plant native to Brazil used to treat some inflammatory states. Regarding efficacy, the maximum inhibition obtained for both ear edema and neutrophil entry was very similar to that produced by topical application of dexamethasone, which prevented the increase in interleukin-1 levels [34]. Medeiros et al. evaluated some of the mechanisms by which α-amyrin exerts its effects against 12-*o*-tetradecanoylphorbol-acetate (TPA)-induced skin inflammation in mice. The topical application of α-amyrin dose-dependently inhibited the increase in prostaglandin E2 (PGE2) induced by TPA. Evaluation of the nuclear factor κB (NF-κB) pathway revealed that topical treatment with α-amyrin is capable of preventing degradation of IκBα, phosphorylation of p65/RelA and activation of NF-κB [35].

It is possible that the *S. dendroideum* extract exerts activity at the core level, and at the level of inflammatory enzymes studied, the effects of the compounds octacosanol and triacontanol [36], the same ones that are present in the extract of *S. dendroideum*, on the activity of the COX and 5-LOX enzymes in vitro. The effects of the addition of these alcohols on the activities of the COX-1, COX-2, and 5-LOX enzymes were evaluated in rat platelet microsomes, rat seminal vesicle microsomes, and rat polymorphonuclear (PMN) preparations, respectively. The addition of octacosanol was ineffective, while triacontanol had a significant effect, it showed competitive, dose-dependent and marked inhibition of activity, which can be reflected in the proliferation and maturation stages in the inflammatory process of scarring through intense inflammatory response [36]. Jurjus et al. reported the effect of β-sitosterol on wounds and burns as the main active ingredient in an ointment, providing hydration to provide healing benefits [37].

The results obtained indicate that the traditional medical use of the sap of *S. dendroideum* in the cicatricial process present in pterygium is fully justified.

## 4. Materials and Methods

### 4.1. Plant Material, Extraction, and Preparation of Extract

Leaves of *S. dendroideum* plants conditioned at 29 ± 4 °C as a greenhouse crop from plants from the community of Las Tazas, Cuautlixco in the state of Morelos, Mexico (18°50′14.3″ N; 98°56′58.44″ W, 1340 masl) were harvested when plants were not flowering. The leaves of *S. dendroideum* D.C. were pressed in a press set (GELGOOG^®^, GG180, Zhengzhou, China) under a pressure of 50 kg/cm^2^. The juice was filtered with cellulose filter paper (20–25 µm) and dried in a lyophilizer (Virtis^®^ Consol 12EL, Cridersville, OH, USA). The final powder concentrate (Pw) was stored at −20 °C in a freezer (General Electric, model FCM 15 SPA WH, Houston, TX, USA). Extractions from 30 g of the Pw were carried out with 2 L of organic solvents of different polarities: dichloromethane, methanol, and water. The dichloromethane, methanol, and aqueous extracts were concentrated under reduced pressure in a Heidolph^®^ rotary evaporator (Heizbad Hei-VAP model, Schwabach, Germany) at the respective boiling temperatures of the solvents and dried using a HETO DRYWINNER model DW3 lyophilizer (Thermofisher Scientific^®^, Waltham, MA, USA).

### 4.2. GC-MS Analysis of Dichloromethane Extract

The methodology described in the literature was used [38]. The chemical composition of dichloromethane extract was analyzed on a gas chromatograph–mass spectrometry (GC-MS) equipped with a quadrupole mass detector in electron impact mode at 70 eV. Volatile compounds were separated onto an HP 5MS capillary column (25 m long, 0.2 mm i.d., with 0.3 µm film thickness). The oven temperature was set at 40 °C for 2 min, then programmed from 40–260 °C at 10 °C/min and maintained for 20 min at 260 °C. Mass detector conditions were as follows: interphase temperature 200 °C and mass acquisition range, 20–550. Injector and detector temperatures were set at 250 and 280 °C, respectively. Splitless injection mode was carried out with 1 µL of each fraction (3 mg/mL solution). The carrier gas was helium at a flow rate of 1 mL/min. Identification of the compounds was performed with the comparison of mass spectra with those of the National Institute of Standards and Technology (NIST, 1.7 Library, Gaithersburg, MD, USA).

### 4.3. Development of Murine Model of Pterygium-Like Eye Lesion

#### 4.3.1. Animals

We used ICR male mice, eight to ten weeks old, weighing 32 to 36 g. Groups with a minimum number of animals (n = 6) were managed to obtain consistent data. The study was approved by the Ethics Committee of the South Biomedical Research Center of the IMSS, on 7 March 2017, with registration number R-2017-1702-13 complying with the Official Mexican Standard NOM-062-ZOO-1999, Technical specifications for the production, care and use of laboratory animals.

#### 4.3.2. Concentration Response Curve of the Effect of TPA on Ocular Damage

A batch of 30 mice was divided into five groups of six individuals. A different dose of TPA was administered intravitreally in three of the groups (0.125, 0.312, and 0.625 mg in 20 microliters of vehicle, a mixture of acetone with water 25:75), the other two groups were administered with vehicles. Under surgical anesthesia (sodium pentobarbital; 55 mg/kg ip), the intravitreous injection was at 0.5 mm from the posterior canthal angle in the subconjunctival area and oriented to the center of the left eye. The 27 G needle penetrated a length of 2 mm and 30° of inclination from the surface plane. The right eye remained intact as a control. Ophthalmic sodium fluorescein strips were moistened with Splash^®^ (Sophia Laboratories, Mexico City, Mexico) artificial tear solution and applied to reveal the corneal lesions through green fluorescence.

#### 4.3.3. Development of Eye Injury

A total of 36 mice was divided into six groups (n = 6). The treatment groups were named as follows:

HC group: Healthy control which received only artificial tears (Splash®, Sophia Laboratories, Mexico City, Mexico)

NegCtrl group: Negative control animals in which TPA lesions were induced at 0.312 mg/eye intravitreally and did not receive treatment.

Dex group: Positive control, which received conventional treatment of dexamethasone 0.1% ophthalmically.

TxExp group: A group of animals administered with 20 mg/mL dichloromethane extract dissolved in vehicle (cyclodextrin and Tween 20 solution).

Veh1 group: The group which was administered the dilution vehicle intravitreally with water and acetone solution 75:25% *v*/*v*, thus confirming that the damage was caused by TPA, and not by acetone.

Veh2 group: The group which received the solution of cyclodextrin and Tween 20 ophthalmically, in order to verify that Tween 20 does not cause alterations in the ocular tissue.

The mice belonging to the NegCrtl, Dex, and TxExp groups received a challenge of 0.312 mg of TPA, under the procedure indicated above. This concentration was established from the concentration response curve. The treatments were administered ophthalmically under the following scheme: 6 applications of 25 µL throughout the day for the first 4 days, then 4 applications of 25 µL throughout the day until completing 14 days of treatment [5].

#### 4.3.4. Pterygium-Like Model Evaluation with Digital Image Analysis (DIA)

Both eyes of the six groups were evaluated using micrographs under 370–404 nm and 400–700 nm (3000 K) light. Micrographs were captured every third day at a constant focal distance, in a capture system of digital photographs consisting of a Velaquin^®^ VE-MC5 digital camera in a Sarasota^®^ stereoscopic microscope in a “white box” system [39]. Lesions of all the left eyes of the groups administered with TPA (NegCtrl, Dex, and TxD) were identified by the characteristic green color of the sodium fluorescein exposed under black light.

Once the micrographs of both eyes of the six groups of mice were captured, the digital image analysis (DIA) was done using Image J V.1.50 software (National Institute of Health, MD, USA). The images were stored in TIFF digital format with a resolution of 2592 × 1944 pixels in 32-bit color. A millimeter scale was placed as a reference for the subsequent calibration of the micrographs by DIA.

To begin the DIA, it was necessary to contrast the lesion of the left eye from the background, defining the lesions as objects using the “Color Threshold” tool using “otsu” as a filter, in the HSB color space, H (55 a110), S (0−255), and B (48–255). Each lesion was separated from the background of the image using the “Clear outside” tool. The images of each lesion were transformed to 8-bit grayscale and subsequently transformed to a black and white image with the “Make binary” tool, to perform the morphometric analysis of the lesion.

To measure the size and shape of the lesions, the “Measure” tool was used. A pixel size of 2.61 × 10^−3^ mm/px was used to measure the perimeter (mm) and the contour (area) of the object (mm^2^) for the surface of the lesion and used to calculate the percent damaged area of the eye with respect to the total area of the left eye (Equation (1)).
(1)$$\%damaged\;area=\frac{damaged\;area\;\left({\mathrm{mm}}^{2}\right)}{total\;area\;\left({\mathrm{mm}}^{2}\right)}\times 100$$

Moreover, the diameter of the feret was measured because, as it is larger, it indicates more elongated shapes, with irregular closure of the lesion in the direction of one side. If it is smaller, it indicates rounder shapes, with a homogeneous closure from the edges towards the center of the lesion.

To determine the shape of the lesions, morphometric descriptors “circularity” and “solidity” were used. In the menu “set measurements” and “shape descriptors”, solidity indicates the convexity of the area as the value got closer to 1. As the circularity value got closer to 0, it indicated elongated shapes.

From the binarized images of the lesions, that is, with their information represented by the values of 0 = black and 255 = white, the “Fractal box count” tool (2, 3, 4, 6, 8, 12, 16, 32, 64) was used to determine the value of the fractal dimension of area (DFA) and perimeter (DFP). The entire eyeball was segmented as an object using the “Crop” tool, the ocular opacity was measured indirectly by the luminosity of the red channel (RGB) and the opacity delta was calculated with Equation (2), separating with “RGB stack” and taking the maximum of the curve from the histogram of the “Threshold” tool.
(2)ΔOpacity=Luminosity of the left eye−Luminosity of the right eye

To determine the vascularization in the left eye, the veins were separated from the rest of the image. The area corresponding to the eyeball was selected and with the tools “Enhance contrast”, “Subtract background” and “split channels”, the colors of the images were separated into red, green, and blue. Subsequently, the image obtained in the green channel was binarized, and the number of pixels that compose it was determined using the plugin “voxel counter”, thus achieving a quantitative evaluation of angiogenesis. The number of pixels was converted to microns, the angiogenesis values were normalized in relation to the healthy state using the average value of the HC group, as in the establishment of Equation (3):(3)$$IA=\frac{µday}{µprom}$$
where: 

µ*day* = µm determined from pixels of each day

µ*prom* = average in µm calculated from the pixels of each day of the basal group

Once the evaluation time was completed, the animals were sacrificed on day 20, and the eyes were removed for the analyses described in the following sections.

#### 4.3.5. Preparation of Eye Tissue for Determination of Cytokine Levels

The eyes were put on a vial with 0.5 mL of phosphate buffer solution (PBS, pH = 7.4) and phenyl-methyl-sulfonyl fluoride (PMFS) at 0.01% dissolved in isopropyl alcohol (Merck, Darmstadt, Germany). Eyes were completely homogenized for 15 s in a homogenizer (Dragon Lab D-500 Pack 1, 10–500 mL). Then, the samples were centrifuged at 14,000 RPM for 5 min. Aliquots of 300 μL were immediately stored at −70 °C for cytokine analyses.

TNF-α and IL-10 cytokine kits were obtained from OptEIA™ ELISA sets (BD Biosciences, Franklin Lakes, NJ, USA) and used following the manufacturer’s instructions. For all cytokines, we added 100 µL of previously prepared o-phenylenediamine (OPD) substrate (one tablet of OPD and one of urea dissolved in 20 mL of distilled water) to each well. This was incubated for 30 min at room temperature in total darkness. A stop solution was added (2N·H_2_SO_4_). The plate was read using a Stat Fax 2100 Spectrophotometer (Awareness Technologies, Bellport, NY, USA) at 450 nm wavelength at 37 °C.

#### 4.3.6. Histological Analysis

Both eyes of the groups that had injuries (NegCtrl, Dex, and ExpTx) and healthy mice without any type of treatment were enucleated and included in paraffin as described in protocols and histological techniques of the University of Vigo [40]. The eyeballs were immersed in 10% formaldehyde and included in paraffin. The fixed tissue was first kept in water, changing every 10 minutes, then progressively dehydrated by sequential periods of 15 min in 50%, 80%, then 100% (*v*/*v*) ethanol solutions, then placed in 50/50 (*v*/*v*) solution of Histochoice^®^ solution (Sigma-Aldrich, Darmstadt, Germany) and ethanol, and finally, in 100% Histochoice^®^. The samples were left immersed in liquid paraffin for 4 h, then placed in inclusion cassettes. The tissue was sectioned with a Leica EM UC7 microtome, stained using the Gomori Trichrome technique that is used to show the increase in collagen deposition associated with the replacement of functional tissue with scar tissue [13]. They were assembled with Entellan mounting medium, and the sections were observed in Nikon Eclipse 80i compound microscope, Japan, at 20×, 40×, and 100×, coupled with a 3C CD-MTI digital camera. Micrographs were captured using Metamorph V 6.1 software (Silicon Valley, CA, USA).

#### 4.3.7. Statistical Analysis

The data were analyzed in IBM SPSS Statistics 25 (Armonk, NY, USA). The anti-inflammatory data were analyzed using ANOVA. Statistical comparisons were evaluated using Dunnet tests when ANOVA *p*-values were below 0.05. Descriptors were compared between groups measured on the same days with Student’s *t*‐test and ANOVA. Cytokines were analyzed by ANOVA tests and post hoc Tukey (n = 6 per treatment group).

## Figures and Tables

**Figure 1 molecules-26-04502-f001:**
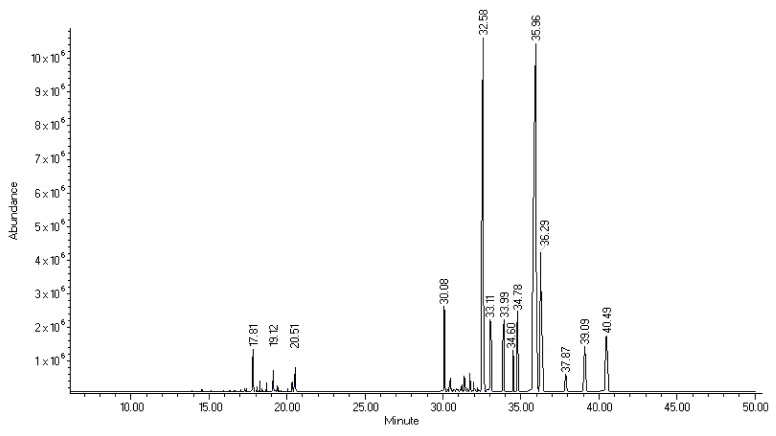
Mass spectrum of DME compounds analyzed by GC-MS. β-Sitosterol (36.29 min), α-tocopherol (33.11 min), α-amiryn (37.87 min), methyl octacosanoate (32.58 min), and in a lower percentage: phytol (17.81), gedunin (30.08 min), hexacosanol (31.85 min), octacosanoic acid methyl ester (33.99 min), α-tocopherol (33.11 min), ciclooctacosane (34.78 min), α-amyrin (37.87 min), and 1-Dotriacontanol (33.99 min).

**Figure 2 molecules-26-04502-f002:**
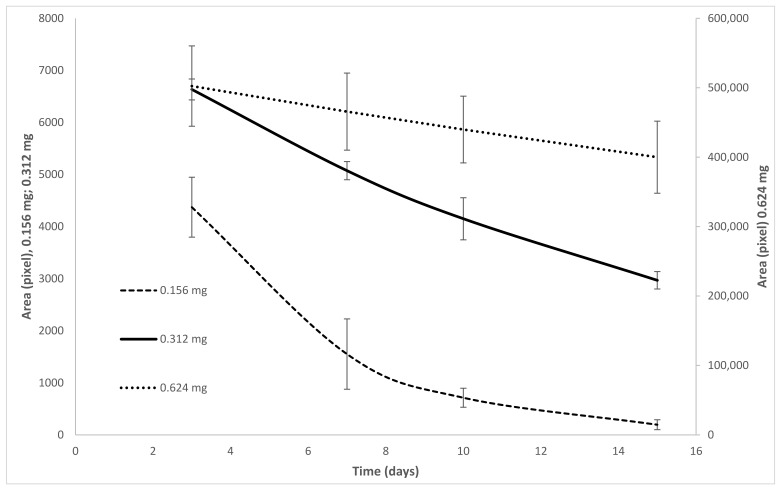
Dose response curve of the daily evolution of the eye injury caused by TPA. Tissue damage was evaluated by the lesion surface revealed by fluorescein. The damaged area was quantified in pixels. The evolution trend of the lesion followed a behavior that was adjusted to an exponential decay curve: y = 5.32 × 10^5^e^−0.019x^ (TPA doses 0.624 mg); y = 8111e^−0.067x^ (TPA doses 0.312 mg); y = 9509e^−0.259x^ (TPA doses 0.156 mg) (n = 6).

**Figure 3 molecules-26-04502-f003:**
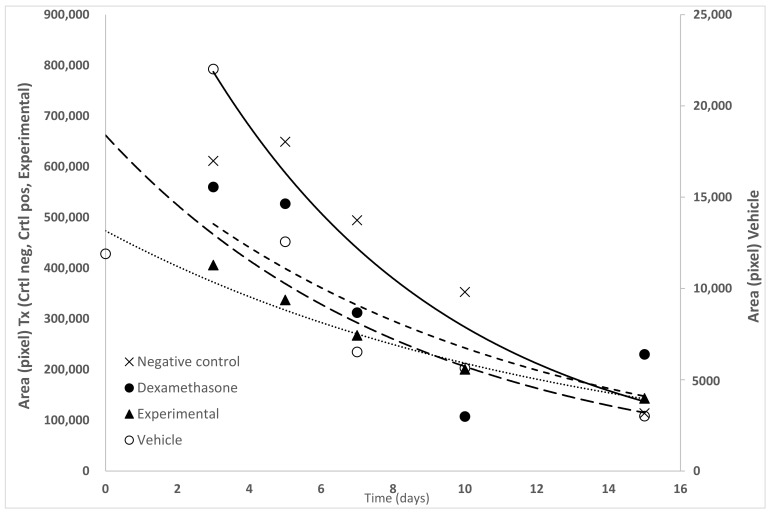
Effect of treatment with *S. dendroideum* extract by ophthalmic application, measuring the evolution of the size of the ocular lesion area, revealed by the application of Figure 2.00 × 10^6^e^−0.174x^ Negative control (× ×), 6.57 × 10^5^e^−0.1x^ Dexamethasone treatment (● ●), 3.45 × 10^5^e^−0.171x^ Vehicle treatment (○ ○), 4.35 × 10^5^e^−0.108x^ Experimental treatment (▲ ▲) (n = 6).

**Figure 4 molecules-26-04502-f004:**
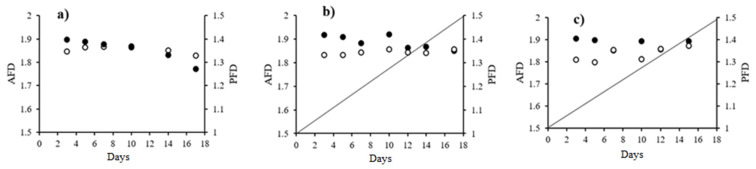
Representation of the perimeter fractal dimension (PFD) (○) and area (AFD) (●) of the treatment groups. Negative control panel (**a**), to AFD y = −0.0073(x) + 1.91 and PFD y = 0.00002(x) + 1.35. Positive control panel (**b**), to AFD y = −0.004(x) + 1.91 and PFD y = 0.001(x) + 1.32. Experimental treatment panel (**c**), to AFD y = −0.003(x) + 1.88 and PFD y = 0.004(x) + 1.275. The value of 2 indicates the area fractal dimension (AFD) and the value 1 is the limit for the perimeter fractal dimension (PFD) (n = 6).

**Figure 5 molecules-26-04502-f005:**
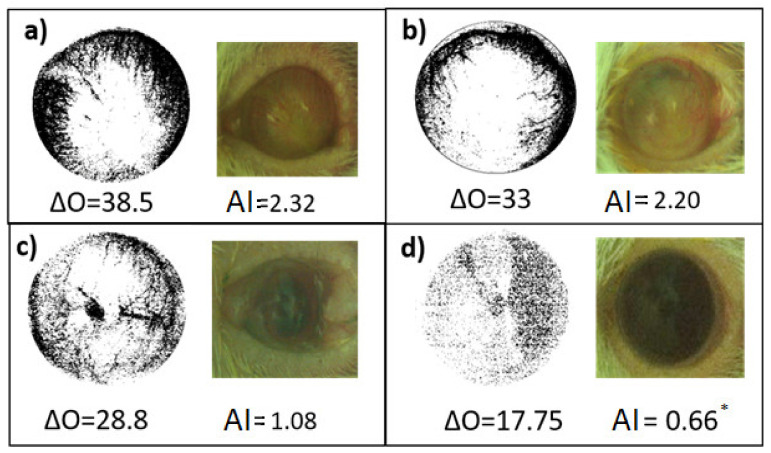
Comparison of the angiogenesis index (AI) and the opacity value (∆O) at the end of the evaluation with respect to the treatment applied. (**a**) Negative Control Group; (**b**) Positive Control Group; (**c**) Experimental Treatment Group; (**d**) Basal Group. ANOVA post Bonferroni test. * *p* < 0.05 is different when compared with negative group (n = 6).

**Figure 6 molecules-26-04502-f006:**
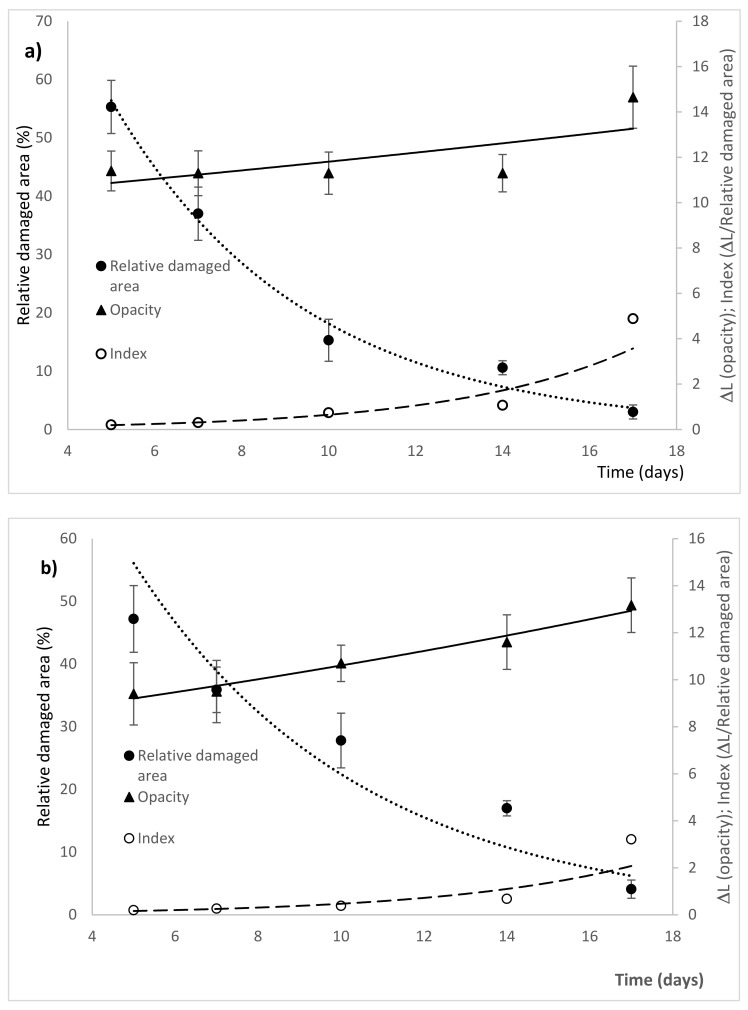
Representation of the time course of changes in relative damaged area, ocular opacity, and the index that relates the opacity and the relative damaged area. Negative control group in panel (**a**), (▲) y = 1173.35e^−0.277x^, (●) y = 10.00e^0.016x^, (○) y = 0.057e^0.243x^. Positive control in panel (**b**), (▲) y = 7.984e^0.028x^, (●) y = 140.09e^−0.183x^, (○) y = 0.057e^0.212x^. Experimental treatment in panel (**c**), (▲) y = 15.48e^−0.058x^, (●) y = 70.68e^−0.115x^, (○) y = 0.219e^0.056x^ (n = 6).

**Figure 7 molecules-26-04502-f007:**
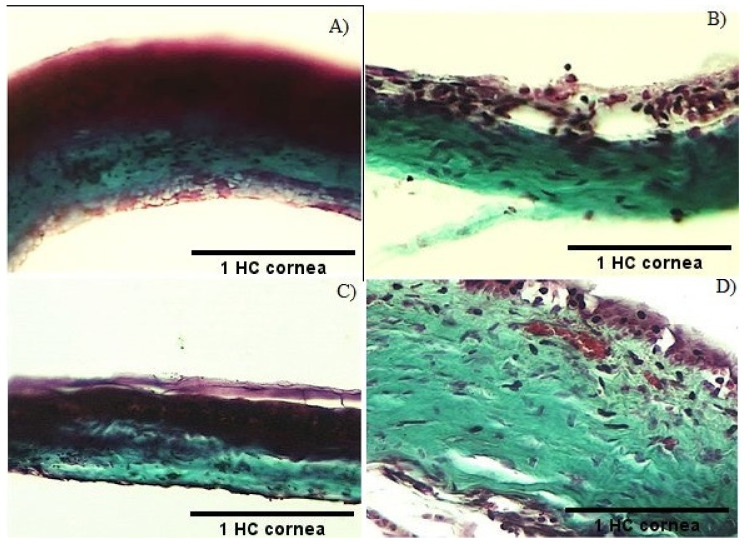
Histological section of cornea stained by Gomori’s trichrome technique from groups that presented injuries. Stained in black is the cell nuclei, in blue the collagen, and in red the muscle endothelium. In (**A**) panel in a healthy cornea, HC group, the intermediate layer called stroma was of the same thickness as the corneal epithelium (20×). The (**B**) panel shows the portion of the cornea belonging to NegCtrl, where the corneal epithelium decreased in thickness with respect to the stroma and presents an apparent decrease in the density of the cell population, with the opening of spaces between cells and detachment of a few (20×). In (**C**) panel in the Dex group, there is detachment of the epithelial layer and apparent opening of spaces in the stroma stained in blue (20×). The TxD group on (**D**) panel presented cell detachment in both the epithelium and the endothelium, and an apparent increase in the density of the cells is observed, evidenced by higher red staining spots on the epithelial surface (40×). Scale bar = 131.0343 pixels is the thickness of the HC cornea and is compared with the other groups.

**Figure 8 molecules-26-04502-f008:**
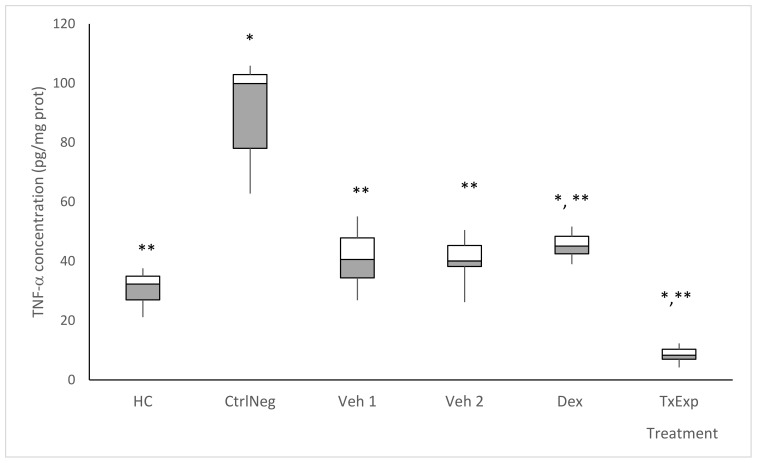
Tissue concentration of Tumor Necrosis Factor α (TNF-α). The treatment groups correspond to the mice without any treatment (HC). The negative control group was administered with TPA. The Veh1 and Veh2 control groups correspond to the TPA dissolution vehicle (acetone) and the treatment dissolution vehicle (cyclodextrin and Tween 20). The positive control received TPA and dexamethasone (0.5%). *S. dendroideum* extract is the experimental treatment. Post hoc Tukey ANOVA n = 6, (**) *p* < 0.05, with respect to the negative control group and (*) *p* < 0.05 with respect to the healthy control group (n = 6).

**Figure 9 molecules-26-04502-f009:**
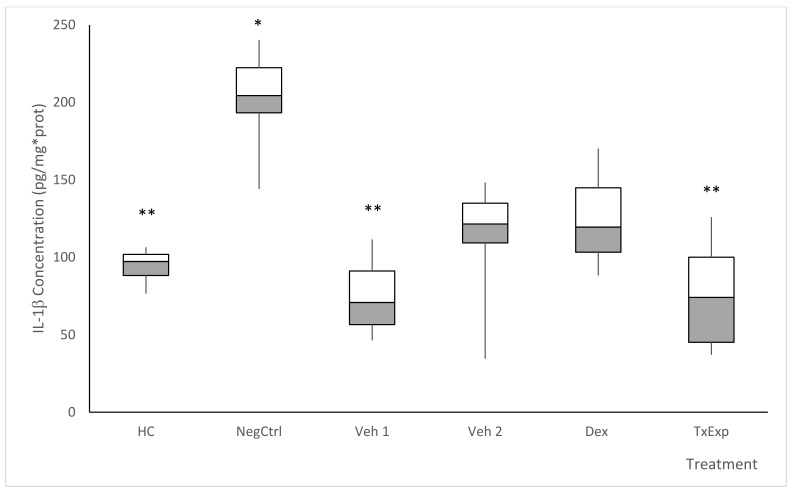
Tissue concentration of interleukin 1β. The treatment groups correspond to the mice without any treatment (HC). The negative control group was administered with TPA. The Veh1 and Veh2 control groups wcorrespond to the TPA dissolution vehicle (acetone) and the treatment dissolution vehicle (cyclodextrin and Tween 20). The positive control received TPA and dexamethasone (0.5%). *S. dendroideum* extract is the experimental treatment. Post hoc Tukey ANOVA, n = 6, (**) *p* < 0.05, with respect to the negative control group and (*) *p* < 0.05 with respect to the baseline control group (n = 6).

**Figure 10 molecules-26-04502-f010:**
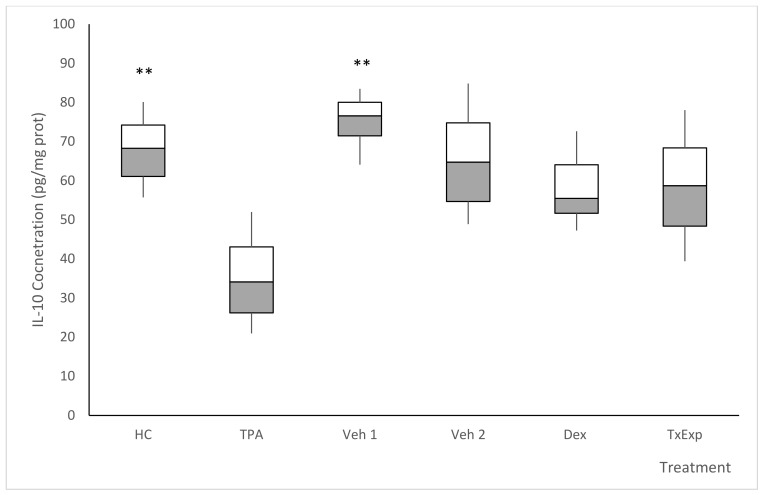
Tissue concentration of interleukin 10. Treatment groups correspond to mice without any treatment (HC). The negative control group was administered with TPA. The Veh1 and Veh2 control groups correspond to the TPA dissolution vehicle (acetone) and the treatment dissolution vehicle (cyclodextrin and Tween 20). *S. dendroideum* extract is the experimental treatment. Post hoc Tukey ANOVA n = 6, (**) *p* < 0.05, with respect to the negative control group.

**Table 1 molecules-26-04502-t001:** Wound size descriptors.

Descriptor	Groups	Initial		Final	
		Mean	SE	Mean	SE
%Injured Aread	NegCtrl	58.16	10.91	26.05 *	9.00
	Dex	39.82	12.22	17.99	6.65
	TxExp	38.76	8.53	30.34	6.41
Perimeter of Injury (mm)	NegCtrl	37.45	2.87	30.25	10.32
	Dex	48.26 ^&^	7.00	20.75 *^,&^	5.40
	TxExp	34.1 ^&^	4.72	35.93 ^&^	3.21
Feret’s Diameter (mm)	NegCtrl	2.71	0.30	1.13 *	0.39
	Dex	2.95	0.39	1.23 *	0.38
	TxExp	2.97	0.34	2.47 *^,&^	0.33

ANOVA post Bonferroni test, * *p* < 0.05 when the initial state is compared with the final one within each group, ^&^
*p* < 0.05 when each group is compared with the NegCtrl (n = 6).

## Data Availability

The data presented in this study are available upon request from the corresponding author. The data is not publicly available because it is not available in any publication.

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
