# Peer review of "Corneal Healing and Recovery of Ocular Crystallinity with a Dichloromethane Extract of Sedum dendroideum D.C. in a Novel Murine Model of Ocular Pterygium"

_molecules, 2021, doi:10.3390/molecules26154502_

Round 1

Reviewer 1 Report

Dear Authors,

The authors are reporting the beneficial effect of an extract of Sedum dendroideum (ESD), in an ocular pterygia mouse model. The treatment of with the ESD decreased the inflammatory reaction, due to the injury, decreased the level of neovascularization over the cornea, and helped to improve the corneal opacity compared to the pterygia cornea. In addition, the ESD treatment shows a better outcome. The use of natural compounds is a very interesting approach because numerous natural compounds proved to be effective in in vitro and in in vivo studies, to treat diseases.

I have some comments about the manuscript.

Major:

  • For each figure, the authors must provide the number of samples used per conditions, because 36 animals split in 6 groups are mentioned but also a minimum of 6 animal per conditions are used. It is not clear. Only table 1 is reporting the n.
  • For the figure 6, a scale will provide a better information than the magnification, because based on the magnification, it seems that the corneal thickness of the Treated group almost double in thickness. Changes in thickness and stromal disorganization should be commented.
  • It seems that reference [X] (g., line 424, 429...) are used as the subject of a sentence. A subject should be used instead of a reference that should be placed at the end of the sentences.
  • Part from line 435 to 446 and part from line from 447 to 458 are the same. This must be corrected. Also, reference 34 or reference 38 are used in these 2 parts. The authors should revise all the references over the manuscript to ensure they are accurate.
  • The English should be revised.
  • Do the authors know natural compounds used in clinics or approved by federal agencies?

Minor:

  • Title: dendroideum is a full one name.
  • Minor editing should be done like correcting the symbol (alpha, beta…)
  • References all over the manuscript are highlighted with colors. This must be corrected.
  • I’m not sure if it is a conversion manuscript problem, but the legend of figure 3 is difficult to read and messed up. This should be corrected.
  • Figure 5: Do the authors mean AI, rather than IA?

Author Response

Reviwer 1

Dear Authors,

The authors are reporting the beneficial effect of an extract of Sedum dendroideum (ESD), in an ocular pterygia mouse model. The treatment of with the ESD decreased the inflammatory reaction, due to the injury, decreased the level of neovascularization over the cornea, and helped to improve the corneal opacity compared to the pterygia cornea. In addition, the ESD treatment shows a better outcome. The use of natural compounds is a very interesting approach because numerous natural compounds proved to be effective in in vitro and in in vivo studies, to treat diseases.

I have some comments about the manuscript.

Major:

  1. For each figure, the authors must provide the number of samples used per conditions, because 36 animals split in 6 groups are mentioned but also a minimum of 6 animal per conditions are used. It is not clear. Only table 1 is reporting the n.

Response:

The number of individuals per group in each condition was 6 (n = 6), since according to the ethics committee of the Centro de Investigación Biomédica del Sur of the Instituto Mexicano del Seguro Social, it is sufficient for our work, because we found statistically significant differences in the results, as well as reproducibility of the model.

  1. For the figure 6, a scale will provide a better information than the magnification, because based on the magnification, it seems that the corneal thickness of the Treated group almost double in thickness. Changes in thickness and stromal disorganization should be commented.

Response:

The magnification of the photographs was corroborated, and that of the TxExp group is appearly to be twice as thick because it is at 40X, at this magnification the details of the organization of the stroma and cell proliferation were recorded more clearly, these changes were confirmated later with LSCM (data not shown). Therefore, the scale was established taking as reference the thickness of the cornea of the HC group and the reference bar is equivalent to a one thickness of HC cornea, which is 131.0343 pixels. Thus it was possible to clearly compare the thickness and dimensions between the corneas. The text mentions the difference in thickness of the negative control group due to the absence of epithelium, since the description of the differences in thickness and cell changes between groups has already been done.

  1. It seems that reference [X] (g., line 424, 429...) are used as the subject of a sentence. A subject should be used instead of a reference that should be placed at the end of the sentences.

Response:

The structure of the sentences that had the reference as a subject was modified. The corresponding authors were located and the reference was moved to the correct place.

  1. Part from line 435 to 446 and part from line from 447 to 458 are the same. This must be corrected. Also, reference 34 or reference 38 are used in these 2 parts. The authors should revise all the references over the manuscript to ensure they are accurate.

Response:

The repeating paragraph that included lines 435 to 446 and 447 to 458, which was the same, was removed.

  1. The English should be revised.

Response:

English was checked and corrected.

  1. Do the authors know natural compounds used in clinics or approved by federal agencies?

Response:

No. The authors are aware of and were guided by the treatment approved by federal health institutions in Mexico, with ophthalmic anti-inflammatory drugs such as Dexamethasone or Indomethacin and finally surgical resection.

Minor:

  1. Title: dendroideum is a full one name.

Response:

dendroideum is part of the full species name. However, the part corresponding to the genus Sedum was edited to S. dendroideum.

  1. Minor editing should be done like correcting the symbol (alpha, beta…)

Response:

The manuscript was reviewed and the symbol α, β was corrected

  1. References all over the manuscript are highlighted with colors. This must be corrected.

Response:

Highlighter was removed from references marked in blue.

  1. I’m not sure if it is a conversion manuscript problem, but the legend of figure 3 is difficult to read and messed up. This should be corrected.

Response:

We edited figure 3 by adding "Time (days)" on the X axis.

  1. Figure 5: Do the authors mean AI, rather than IA?

Response:

The IA letters were changed to AI.

Submission Date

26 May 2021

Date of this review

13 Jun 2021 03:56:41

Reviewer 2 Report

Please explain the difference between the Pterygium and "a corneal lesion was caused by intravitreal injection of TPA". Can this model represent the tissue changes and pathophysiological changes of Pterygium? And explain its shortcomings.

Author Response

Reviwer 2

  1. Please explain the difference between the Pterygium and "a corneal lesion was caused by intravitreal injection of TPA". Can this model represent the tissue changes and pathophysiological changes of Pterygium? And explain its shortcomings.

Response:

The main difference with human pterygium is the absence of scar tissue on the cornea, this due to the structure of the mouse eye, where the proliferation of fibroblasts and connective tissue does not occur as in humans. However, there is greater proliferation and cell density in the injured cornea, compared to a healthy eye, as occurs in the human pterygium, this was verified with digital analysis using the LSCM and ESEM tools (data not shown).

Submission Date

26 May 2021

Date of this review

29 Jun 2021 21:10:30

Reviewer 3 Report

  1. Title: Please correct ''Sedum den-droideum'' to ''Sedum dendroideum'' and  ''pterygia'' to ''pterygium''.
  2. Line 27, 30-31, 90: Please correct IL-1@, TNF-@.
  3. Line 92: Please correct the italic font of ''Sedum dendroideum'' through the full manuscript.
  4. Line 104: ''CG'' change to ''GC''.
  5. Line 160: Please correct the Fig. 3 legend.
  6. Please correct [(**) p <0.05 and (*) p <0.05] to [[(**) p <0.01 and (*) p <0.05] through the full manuscript.
  7. Please add animal welfare and provide IACUC approval number in the M&M.
  8. Please cited thr recent references in this manuscript.

Author Response

Reviwer 3

  1. Title: Please correct ''Sedum den-droideum'' to ''Sedum dendroideum'' and ''pterygia'' to ''pterygium''.

Response:

Pterygia was changed to pterygium.

  1. Line 27, 30-31, 90: Please correct IL-1@, TNF-@.

Response:

Corrected the @ in lines 27, 30, 31 and 90, for the correct symbol.

  1. Line 92: Please correct the italic font of ''Sedum dendroideum'' through the full manuscript.

Response:

The italic font was changed

  1. Line 104: ''CG'' change to ''GC''.

The CG letters where changed to GC.

  1. Line 160: Please correct the Fig. 3 legend.

Response:

The legend was changed

  1. Please correct [(**) p <0.05 and (*) p <0.05] to [[(**) p <0.01 and (*) p <0.05] through the full manuscript.

Response:

Regarding the observation of modifying [(*) p <0.05 and () p <0.05] to [[(*) p <0.01 and () p <0.05], we worked with p <0.05 because it is one of the important criteria to have statistically significant differences in the evaluation of biological phenomena, so that p <0.01 is not the most suitable for this purpose.

  1. Please add animal welfare and provide IACUC approval number in the M&M.

Response:

In materials and methods (M&M) it is already mentioned that it was worked under the approval of an ethics committee of the Centro de Investigación biomédica del Sur del Insituto Mexicano del Seguro Social, we are governed by the Official Mexican Standard NOM-062-ZOO-1999, Technical specifications for the production, care and use of laboratory animals. That is why the IACUC number was not presented.

  1. Please cited thr recent references in this manuscript.

Response:

Regarding the observation on the use of more current references, for the purposes of the work, the oldest references sufficiently fulfill the function of defining and helping to limit the work, since the most basic in terms of antecedents, is presented in the cited articles.

Submission Date

26 May 2021

Date of this review

24 Jun 2021 03:18:51

Round 2

Reviewer 1 Report

Dear authors,

I have no additional comments.

Sincerely.

Author Response

No comment

Reviewer 2 Report

The authors have made explanations and amendments, We suggest that difference between this model and human disease (pterygium and corneal lesion) can be explained in the manuscript.

Author Response

Comments and Suggestions for Authors

The authors have made explanations and amendments, We suggest that difference between this model and human disease (pterygium and corneal lesion) can be explained in the manuscript.

Response:

A paragraph was included in the discussion in which the differences between the animal model used and the human pterygium are exposed.

Reviewer 3 Report

No

Author Response

No comment